# Early-Onset Versus Late-Onset Preeclampsia in Bogotá, Colombia: Differential Risk Factor Identification and Evaluation Using Traditional Statistics and Machine Learning

**DOI:** 10.3390/biomedicines13081958

**Published:** 2025-08-12

**Authors:** Ayala-Ramírez Paola, Mennickent Daniela, Farkas Carlos, Guzmán-Gutiérrez Enrique, Retamal-Fredes Eduardo, Segura-Guzmán Nancy, Roca Diego, Venegas Manuel, Carrillo-Muñoz Matias, Gutierrez-Monsalve Yanitza, Sanabria Doris, Ospina Catalina, Silva Jaime, Olaya-C. Mercedes, García-Robles Reggie

**Affiliations:** 1Human Genetics Institute, Faculty of Medicine, Pontificia Universidad Javeriana, Bogotá 110231, Colombia; 2Red Iberoamericana de Alteraciones Vasculares en Trastornos del Embarazo (RIVATREM), Chillán 3780000, Chile; 3Departamento de Ciencias Básicas y Morfología, Facultad de Medicina, Universidad Católica de la Santísima Concepción, Concepción 4090541, Chile; cfarkas@ucsc.cl; 4Grupo Inicial de Investigación en Tecnología e Innovación en Salud para el Bienestar de las Personas (VITALIS), Universidad Católica de la Santísima Concepción, Concepción 4090541, Chile; 5Departamento de Bioquímica Clínica e Inmunología, Facultad de Farmacia, Universidad de Concepción, Concepción 4070386, Chile; 6Tecnología Médica con Mención en Imagenología y Física Médica, Facultad de Medicina, Universidad Católica de la Santísima Concepción, Concepción 4090541, Chile; 7Magíster en Ciencias Biomédicas, Facultad de Medicina, Universidad Católica de la Santísima Concepción, Concepción 4090541, Chile; 8Nutrición y Dietética, Facultad de Medicina, Universidad Católica de la Santísima Concepción, Concepción 4090541, Chile; 9Research Seedbed in Perinatal Medicine, Faculty of Medicine, Pontificia Universidad Javeriana, Hospital Universitario San Ignacio, Bogotá 110231, Colombia; 10Department of Obstetrics and Gynecology, Faculty of Medicine, Pontificia Universidad Javeriana, Hospital Universitario San Ignacio, Bogotá 110231, Colombia; 11Department of Pathology, Faculty of Medicine, Pontificia Universidad Javeriana, Hospital Universitario San Ignacio, Bogotá 110231, Colombia; olaya.m@javeriana.edu.co; 12Department of Physiological Sciences, Faculty of Medicine, Pontificia Universidad Javeriana, Bogotá 110231, Colombia

**Keywords:** late-onset preeclampsia, early-onset preeclampsia, hypertension in pregnancy, Latin America, risk factors, traditional statistics, machine learning, artificial intelligence

## Abstract

**Background/Objectives:** Preeclampsia (PE) is a major cause of maternal and perinatal morbidity and mortality, particularly in low- and middle-income countries. Early-onset PE (EOP) and late-onset PE (LOP) are distinct clinical entities with differing pathophysiological mechanisms and prognoses. However, few studies have explored differential risk factors for EOP and LOP in Latin American populations. This study aimed to identify and assess clinical risk factors for predicting EOP and LOP in a cohort of pregnant women from Bogotá, Colombia, using traditional statistics and machine learning (ML). **Methods:** A cross-sectional observational study was conducted on 190 pregnant women diagnosed with PE (EOP = 80, LOP = 110) at a tertiary hospital in Bogotá between 2017 and 2018. Risk factors and perinatal outcomes were collected via structured interviews and clinical records. Traditional statistical analyses were performed to compare the study groups and identify associations between risk factors and outcomes. Eleven ML techniques were used to train and externally validate predictive models for PE subtype and secondary outcomes, incorporating permutation-based feature importance to enhance interpretability. **Results:** EOP was significantly associated with higher maternal education and history of hypertension, while LOP was linked to a higher prevalence of allergic history. The best-performing ML model for predicting PE subtype was linear discriminant analysis (recall = 0.71), with top predictors including education level, family history of perinatal death, number of sexual partners, primipaternity, and family history of hypertension. **Conclusions:** EOP and LOP exhibit distinct clinical profiles in this cohort. The combination of traditional statistics with ML may improve early risk stratification and support context-specific prenatal care strategies in similar settings.

## 1. Introduction

Preeclampsia (PE) is a heterogeneous and multisystemic disorder specific to pregnancy, characterized by new-onset hypertension, which appears clinically after the twentieth week of gestation, accompanied by new-onset proteinuria or, otherwise, damage to target organs within the framework of an exaggerated systemic inflammatory process [1,2]. The American College of Obstetricians and Gynecologists considers arterial hypertension and proteinuria as classic criteria for the diagnosis of PE. However, it recommends that women with gestational hypertension (defined as systolic pressure blood pressure ≥ 140 mmHg, or diastolic blood pressure ≥ 90 mmHg, or both, on two occasions at least 4 h apart after 20 weeks of gestation in a woman with a previously normal blood pressure) in the absence of proteinuria are diagnosed with PE if they exhibit any of the following characteristics: thrombocytopenia (platelet count per liter < 100,000); impaired liver function as indicated by abnormally high blood concentrations of liver enzymes (up to double the upper limit of their normal concentrations); severe persistent epigastric or right upper quadrant pain not explained by alternative diagnoses; renal insufficiency (serum creatinine concentration > 1.1 mg/dL or a doubling of the serum creatinine concentration in the absence of another kidney disease); pulmonary edema; or new headache that does not respond to medication and is not explained by alternative diagnoses or visual symptoms [1].

PE is one of the main causes of maternal and fetal morbidity and mortality in Colombia. In 2023, hypertensive disorders accounted for the highest rates of extreme maternal mortality in Colombia, with 38.2 cases per 1000 live births [3]. The main cause of maternal death in Colombia is hypertensive disorders of pregnancy with a mean frequency of 20.4% of the last 3 years [4]. A Colombian study of the care of women with PE/eclampsia identified the inopportune diagnosis of the disease as the third factor affecting the quality of care by 18.5% [5], suggesting that early clinical identification and identification of the disease is a challenge for obstetrician-gynecologists or general practitioners who perform prenatal check-ups.

Two subtypes of PE have been determined according to the time of the onset of the disease: early-onset PE (EOP) and late-onset PE (LOP). The first one develops before 34 gestational weeks and the second during or after week 34 of gestation. Both share diagnostic criteria and some etiological characteristics, but differ with respect to risk factors, and lead to different outcomes, so the two types of PE should be treated as distinct entities from an etiological and prognostic point of view [2]. On the other hand, it is also assumed that the pathophysiology of the two disorders is different; EOP is considered a fetal disorder, associated with an underlying placental abnormality, where the typical characteristic is the dysfunctional pathway of remodeling of the uterine spiral arteries given by the defective trophoblastic invasion contributing to hypoperfusion, ischemia, and placental hypoxia, while LOP is considered a maternal disorder where placentas are generally normal and the problem arises from the interaction between a presumably normal placenta and maternal factors plagued with endothelial dysfunction, making them susceptible to microvascular damage. LOP seems to be a decompensate response to the oxidative stress in the placenta caused by dysfunctional maternal endothelium. Endothelial dysfunction, which is one aspect of a systemic maternal inflammatory response, may result in generalized vasoconstriction and reduced blood flow to multiple organs, including the heart, kidneys, and brain [2,6].

A number of studies have described the associated maternal morbidities, birth outcomes, and clinical and laboratory characteristics of the two types of PE, and have concluded that the early-onset disease is more severe and has greater association with eclampsia, hemolysis, elevated liver enzymes, low platelet count (HELLP) syndrome, multisystemic failure, intrauterine growth restriction (IUGR), and small fetuses for gestational age (SGA), with consequent increased maternal and fetal morbidity and mortality. In contrast, late-onset cases tend to be more “benign”, without presenting severe conditions on most occasions and newborns tend to have adequate weight or are large for gestational age [2,7,8,9]. Even though these differences are known, it is still clinically challenging to predict if a particular pregnant woman will present EOP or LOP. Literature shows attempts to address this issue, based both on traditional statistics [2,10,11,12,13] and machine learning modelling [14,15,16,17]. Such articles identify differential risk factors between EOP and LOP, and some test them for EOP and LOP prediction with promising results; however, most of these studies are based on Asian or European population data. Furthermore, the few Colombian articles that evaluate differences between EOP and LOP do not perform any validation of the identified risk factors, nor do they test them as predictors of these conditions by classification metrics [18,19].

Bogotá is the capital and the largest city in Colombia, as well as the third highest capital of South America, at an average of 2640 m above sea level. The peculiarity of this tropical region lies in the multi-ethnic origin of its inhabitants and most of the population is mestizo [20]. This particularity should be considered when identifying risk factors for PE, as well as when developing new predictive models for EOP and LOP, since different populations have different biological, environmental, and cultural characteristics that may favor or disfavor the occurrence of PE. For instance, the Colombian health system has barriers to the accessing of maternity services, directly affecting the quality of care and increasing PE risk factors in comparison to other populations [5]. Notably, there is a lack of studies on PE in the Latin American population, particularly in countries other than Brazil and Mexico [21], which highlights the need to identify and test risk factors particular to the Colombian population in the context of PE prediction. Therefore, the aim of this study was to identify and evaluate differential risk factors between EOP and LOP in women from Bogotá, Colombia, using traditional statistics and machine learning.

## 2. Materials and Methods

### 2.1. Patients

An analytical cross-sectional observational study of cases was carried out. The study population included deliveries between July 2017 and November 2018 from Hospital Universitario San Ignacio in Bogotá, Colombia. A member of the medical staff conducted a personal interview with the pregnant women and other family members at the time of finding a case. The information collected was recorded in a RedCap web-based format. The interviews took place while the mother was hospitalized for delivery care and lasted 15 min approximately. All women whose pregnancies had been included in the case or control group were invited to complete an in-depth PE/eclampsia risk factor survey and to allow the researchers access to their clinical history until they were discharged from Hospital Universitario San Ignacio. Participation in PE screening and subsequent monitoring was entirely voluntary. Between July 2017 and November 2018, 236 women developed PE but only 190 qualified for the study; the other 21 women were not available for interview and 25 women were not included in this analysis because they had incomplete data. The Institutional Research and Ethics Committee of the Faculty of Medicine, Pontificia Universidad Javeriana—Hospital Universitario San Ignacio approved this study and written informed consent was obtained from all study participants before interview.

### 2.2. Variable Definitions and Main Outcomes

PE was defined according to the Task Force on Hypertension in Pregnancy [22]. EOP was defined as PE diagnosed before 34 weeks of gestation and LOP was defined as PE diagnosed during or after 34 weeks of gestation. Eclampsia was defined as the presence of unexplained generalized seizures in patients with PE. For the estimation of gestational age at diagnosis, the early ultrasonogram was used. The gestational age at delivery was calculated according to the Ballard score. HELLP syndrome was defined as the development of hemolysis, elevated liver enzymes, and low platelet count.

The following data were collected: Demographics and clinical findings: gestational age at delivery, maternal age, socioeconomic index according to the Colombian socioeconomic stratification, marital status, maternal education level, time in relationship with the infant’s father, and occupation during pregnancy. Pre-pregnancy maternal characteristics: body mass index (BMI), parity or number of previous live births, age at menarche, number of previous abortions, primigravity (yes/no), primipaternity (yes/no), and prior PE (yes/no). Antenatal characteristics: chronic hypertension (yes/no), allergy (yes/no), migraine (yes/no), hypothyroidism (yes/no), cigarette exposure prior to pregnancy and during the first trimester (yes/no), and alcohol consumption prior to pregnancy and during the first trimester (yes/no). Maternal family history: according to family history (until second-degree relatives) the following data were collected: family history of PE (yes/no), family history of IUGR (yes/no), family history of cardiovascular disease (yes/no), family history of obit or perinatal death (yes/no), family history of spontaneous abortion (yes/no), family history of preterm delivery (yes/no), family history of diabetes (yes/no), family history of cancer (yes/no), and family history of hypertension (yes/no). Newborn outcomes: sex, weight, congenital anomalies in the newborn (yes/no), and IUGR–small for gestational age (SGA) fetus/newborn (yes/no). Other pregnancy outcomes: type of delivery. Perinatal death was defined as the death of either the fetus or of the newborn between the 28th week of pregnancy, or birth weight ≥ 500 g, and the first week of life (7 days); an IUGR–SGA fetus/newborn was considered to be one with a birth weight in the lower 10th percentile of previously published normal curves [23]. Hereafter, the term IUGR will be used in place of IUGR–SGA.

In this study, some of the aforementioned data were considered as outcomes. The primary outcome was the onset of PE, i.e., early- or late-onset. This outcome is categorical. In addition, 5 categorical and 2 continuous secondary outcomes were assessed, respectively: vital status of the newborn; presence or absence of malformations in the newborn; presence or absence of IUGR; type of delivery, i.e., cesarean section or eutocic delivery; presence or absence of eclampsia or HELLP syndrome; gestational age at delivery; and newborn weight. Every other variable recorded was used as a potential risk factor or predictor for such outcomes.

### 2.3. Traditional Statistical Analysis

Nominal categorical variables were compared by Fisher’s exact test when the variable comprised two categories, and the Fisher–Freeman–Halton test when it included more than two categories. Nominal categorical variables included marital status, sex of the newborn, occupation, all personal and family history of disease (including pre-pregnancy and first trimester information), primigravidity, primipaternity, vital status of the newborn, type of delivery, congenital malformations, IUGR, eclampsia/HELLP syndrome, and preeclampsia onset. Among these, “occupation” was the only variable analyzed using the Fisher–Freeman–Halton test. Ordinal categorical variables—namely, education level and socioeconomic status—were compared by the Cochran–Armitage exact test for trend. Both nominal and ordinal categorical variables were summarized as percentage (proportion). The normality of numerical variables was evaluated by the Shapiro–Wilk test. None of the features analyzed had normal distribution. Not-normally distributed numerical variables were compared by the Mann–Whitney test and summarized as median (interquartile range). The numerical variables included maternal age, age at menarche, BMI, time in relationship with the infant’s father, number of abortions, number of pregnancies, number of sexual partners, gestational age at delivery, and newborn weight. *p* values less than 0.05 were considered statistically significant. In addition, a correlation matrix including all predictors and outcomes was computed based on Pearson’s correlation coefficients. These analyses were performed using GraphPad Prism 10.3.1 (GraphPad Software Inc., San Diego, CA, USA) and Python 3.10.18.

### 2.4. Machine Learning Analysis

To develop machine learning models, clinically meaningful outcomes were selected as prediction targets. The selection of input and target variables was carefully structured to prevent redundancies and overfitting of our models. Based on clinical relevance and prior literature, 8 of the 38 features included in the dataset were considered as targets for our machine learning models: PE subtype, newborn vital status, newborn malformations, IUGR, type of delivery, eclampsia/HELLP syndrome, gestational age at delivery, and newborn weight. These clinical outcomes were exclusively used as prediction targets and were systematically excluded from the input feature set. A detailed summary of the variables used as predictors or targets across the different machine learning models is provided in Appendix A.

The full dataset was loaded in Python 3.10.18. using a semicolon-delimited CSV file, with the “id” column set as the index, and subsequently underwent autoscaling (mean-centering and scaling to unit variance) to standardize feature distributions. The dataset was then split into training (80%) and external validation (20%) subsets for each outcome and class, using a fixed random seed (random_state = 7), thereby preserving reproducibility and enabling consistent assessment of model generalizability on unseen data.

For categorical outcomes, a total of nine classification algorithms were evaluated, with their training parameters stated in parentheses: logistic regression (LogReg; using the Newton–Cholesky solver and max_iter = 100), linear discriminant analysis (LDA; default priors), Gaussian naive Bayes (GNB; default priors), k-nearest neighbors (KNN; n_neighbors = 5), decision tree (DecTree; default priors), random forest (RF; bootstrap = False), gradient boosting classifier (GradBoost; max_depth = 5), support vector machine (SVM; probability = True), and a simple multilayer perceptron (MLP; hidden_layer_sizes = 100, max_iter = 300). All classifiers employed random_state = 7 unless otherwise stated. For each outcome, model performance was assessed via macro-averaged recall—where the recall is computed for each class independently and then averaged, so every class has equal influence on the final metric—even in cases of imbalance.

For continuous outcomes such as gestational age and newborn weight, the pipeline incorporated two regression algorithms, for which training parameters are stated in parentheses: a random forest regressor (RF; n_estimators = 100) and a gradient boosting regressor (GradBoost; n_estimators = 100), each with random_state = 7. In this case, the root mean squared error (RMSE) served as the main performance metric, given its measure of how far predictions deviate from true values on average. The goodness of fit was assessed by the mean squared error (MSE), mean absolute error (MAE), and the coefficient of determination (R^2^) metrics.

Following model training with the complete set of predictors, variable selection was conducted. Once the full dataset training had finished, permutation-based feature importance was computed on the held-out validation set (20% split) for each predictive task. The best model for each outcome—defined by the highest macro-recall among classifiers or the lowest RMSE among regressors—supplied a ranked list of feature importances. Two complementary criteria were then applied: every predictor whose importance exceeded 0.02 was kept, and the top quartile (top 25%) of the ranked list was also retained, with the threshold rounded up to the nearest integer. The union of features selected under both criteria across all outcomes resulted in a reduced set of variables. This subset maintained all predictors that made an important contribution to model performance, while excluding less informative features. Final models were trained and validated using this reduced feature set.

The entire analysis pipeline, including results, was implemented in Python 3.10.18. and is publicly accessible at: https://github.com/cfarkas/preeclampsia_ml (accessed on 30 July 2025).

## 3. Results

### 3.1. Population Characterization by Univariate Analysis

Table 1 shows how risk factors and secondary outcomes behave according to the timing of onset of PE. The group of women with EOP is characterized by having a higher educational level, a higher frequency of history of arterial hypertension, and a lower frequency of history of allergy, than the group of women with LOP. As expected, the EOP group is also characterized by a lower gestational age at delivery, newborns with lower weight, and a higher proportion of cesarean sections relative to eutocic deliveries than the LOP group.

Table 2 displays the risk factors that are significantly different in the case of the categorical secondary outcomes studied. The group of pregnant women with a live newborn is characterized by a history of a greater number of pregnancies and a lower frequency of primigravity than those with a stillbirth. On the other hand, the group of pregnant women whose newborn has malformations is characterized by younger age, lower socioeconomic status, and a history of fewer pregnancies than those whose newborn does not have malformations. In the case of IUGR, two variables are statistically significant: history of personal or family IUGR and family history of preterm delivery. The group of pregnancies with IUGR is characterized by a higher frequency of both antecedents compared with those without IUGR. The group of women with cesarean section is characterized by older age than the group of women with eutocic delivery. Finally, the group of women with eclampsia or HELLP syndrome did not differ from the group of women without eclampsia or HELLP syndrome in terms of the characteristics analyzed.

Figure 1 displays a correlation matrix illustrating the relationships among various risk factors and perinatal outcomes included in the study. This matrix was derived from the full correlation matrix—constructed using the entire set of predictors and target variables—to improve clarity and readability. Only variables with correlation coefficients equal to or greater than |0.12| were included. The complete matrix is displayed as Appendix A. Positive correlations were observed between the onset of PE and both neonatal birth weight and gestational age at delivery. In addition, a strong positive correlation was found between neonatal birth weight and gestational age at delivery, indicating—as expected—that higher gestational age is associated with greater birth weight. A positive correlation was also noted between PE onset and type of delivery; in particular, LOP was positively associated with eutocic delivery, whereas EOP showed a positive correlation with cesarean section. These associations are consistent with clinical practice, as EOP typically requires cesarean delivery due to its greater severity and increased maternal–fetal risk. In contrast, LOP tends to be less severe and occurs nearer to term, often allowing for spontaneous vaginal delivery. Notably, a negative correlation was found between the onset of PE and a personal history of hypertension, which is consistent with the association identified between EOP and personal history of hypertension in the univariate analysis. Overall, the matrix provides an integrated view of co-occurrence patterns among clinical characteristics and outcomes, offering valuable insight for the development of predictive models and early screening strategies.

### 3.2. Outcome Prediction by Machine Learning

To predict the different outcomes considered, varied machine learning models were trained and externally validated. In a first attempt, all the risk factors were evaluated as predictors. Appendix A present the predictive power of such classification and regression machine learning models, respectively.

Then, the risk factors that contributed the most to the predictive performance of these models were selected using a permutation-based feature importance approach, which—together with other technical aspects of the training process—is described in detail in Section 2.4. They included personal history of hypertension, personal and family history of diabetes, family history of cardiovascular disease, primigravidity, primipaternity, personal history of allergy, number of sexual partners, number of pregnancies, personal history of migraine, occupation, cigarette consumption before and at the first trimester of pregnancy, age at menarche, education level, BMI, family history of cancer, personal history of hypothyroidism, socioeconomic level, family history of hypertension, number of abortions, alcohol consumption before and at the first trimester of pregnancy, family history of preterm birth, personal history of PE, personal and family history of IUGR, family history of obit or perinatal death, and family history of PE. These risk factors were used to train and externally validate a new set of classification and regression models, aiming to predict the primary and secondary outcomes considered here.

Figure 2 summarizes the classification performance of different models for the prediction of categorical outcomes, using the subset of selected high-ranking features. The recall corresponds to the average between the sensitivity and the specificity of the model. Overall, this metric showed a slight improvement after feature selection, as evidenced by comparing these results with those presented in Appendix A, which displays the recall obtained using the full set of predictors. For each categorical outcome, the model yielding the highest macro-averaged recall was identified as the best-performing classifier. The corresponding feature importance for these models was then computed and is displayed in Figure 3.

Figure 2 shows that the best-performing model for predicting PE onset was obtained using LDA and achieved a recall of 0.71. Figure 3A highlights the variables that drive the LDA model, with the most influential contributors being educational level, family history of obit, number of sexual partners, primipaternity, and family history of hypertension. Furthermore, as shown in Figure 2, the best-performing secondary classifiers were for newborn vital status (GNB, recall = 0.78) and eclampsia/HELLP syndrome (LogReg, recall = 0.62). Delivery type (KNN, 0.53), newborn malformations (GNB, 0.58), and IUGR (GradBoost, 0.56) models reached more modest recalls. The key predictors for these five models are depicted in Figure 3B–F.

Finally, models were developed for continuous outcomes such as gestational age at delivery and newborn weight, but their regression metrics were poor (best R^2^ < 0.25), indicating that additional clinical or biochemical variables will be required to produce reliable numeric predictions. Notably, the results obtained using the complete set of input features, presented in Appendix A, did not substantially improve after applying feature selection, for which results are shown in Appendix A.

## 4. Discussion

This study offers a thorough clinical characterization and differential risk factor evaluation of EOP and LOP among pregnant women in Bogotá, Colombia, employing both traditional statistical methods and machine learning techniques. Our findings reinforce the understanding that EOP and LOP are distinct clinical entities with differing etiopathogenic mechanisms and prognostic implications, aligning with existing literature [2,6].

EOP was correlated with higher maternal educational attainment and a more prevalent history of chronic hypertension, whereas a history of allergies was more common among women with LOP. Although higher education is typically viewed as protective for maternal health, its association with EOP in our cohort may indicate variations in health-seeking behaviors, access to tertiary care facilities, or referral practices within the Colombian healthcare system. The significant link between chronic hypertension and EOP supports the idea that EOP features a more pronounced vascular and placental component [6]. In contrast, the higher incidence of allergic conditions in LOP patients may indicate immune-mediated mechanisms involved in this subtype. It has been previously reported that allergies are associated with an increased risk of PE [24].

As anticipated, women experiencing EOP faced poorer perinatal outcomes, including significantly lower gestational age at delivery, reduced newborn weight, and elevated cesarean section rates. These results corroborate previous studies that have described EOP as being associated with greater severity and frequently linked to IUGR, prematurity, and adverse neonatal outcomes [25].

Among the machine learning techniques assessed, LDA demonstrated the best overall efficacy in differentiating between the two types of PE syndromes with a recall rate of 0.71. Key predictive variables identified included educational level, family history of perinatal mortality, number of sexual partners, primipaternity, and family history of hypertension. These findings suggest that integrating machine learning techniques into clinical workflows could enhance early identification of PE subtypes—which would be especially valuable in resource-limited settings where laboratory diagnostic access is constrained [26,27].

Secondary outcomes such as neonatal death, congenital malformations, IUGR, delivery type, and eclampsia/HELLP syndrome were also evaluated. This study highlights the significance of various risk factors in predicting pregnancy complications among women with preeclampsia, using multiple machine learning models. For example, intrauterine growth restriction was found to be closely associated with family history of preterm birth [28], BMI [29], age at menarche [30], cigarette consumption before [31] and during the first trimester of pregnancy [32], and socioeconomic level [33]. The prediction of delivery type was influenced by familial reproductive factors such as a family history of obstetric complications and preterm birth [34]. Concerning neonatal vital status, significant predictors included family history of hypertension, which has previously been associated with severe pregnancy complications that can result in the death of the newborn [35], allergies [36], and pregestational smoking. Neonatal malformations were predominantly linked to maternal hypothyroidism [37] and a family history of obstetric issues [38]. Lastly, severe complications such as eclampsia or HELLP syndrome were correlated with social determinants—including occupation and socioeconomic status—as well as prematernity and primigravidity [39]. These findings emphasize the importance of a comprehensive approach that integrates clinical, personal, and contextual factors in managing care for women affected by preeclampsia.

While predictive performance for continuous outcomes like gestational age at delivery and newborn weight was limited—likely due to sample size constraints or measurement variability—the overall applicability of machine learning in identifying risk profiles for categorical outcomes shows promise warranting further investigation across larger multicenter cohorts.

This study presents several strengths, including the integration of traditional statistical analysis with machine learning techniques to comprehensively evaluate risk factors and perinatal outcomes in Latin American pregnant women with EOP and LOP. The well-defined cohort, extensive clinical characterization, and the use of external validation for predictive models enhance the robustness and clinical relevance of the findings. However, the study also has limitations. The sample size, while respectable, may be insufficient to fully capture the complexity of rare outcomes, and the single-center design limits the generalizability of the results to broader populations. Additionally, the modest predictive performance for continuous outcomes such as gestational age at delivery and newborn weight suggests potential challenges related to data variability or sample size constraints. The reliance on self-reported information for certain variables, the absence of biomarker data, and the presence of class imbalance may also have introduced biases and limited the predictive power of the models. Future multicenter studies incorporating biomolecular markers are recommended to further validate and extend these findings.

## 5. Conclusions

This study provides a detailed clinical characterization of EOP and LOP in a Colombian cohort, highlighting differential risk factor profiles and perinatal outcomes associated with each subtype. By integrating traditional statistical methods with machine learning approaches, we demonstrated the potential of clinical and sociodemographic variables to predict the onset and severity of PE. Notably, this is one of the first studies in Latin America to apply such combined methodologies in the context of hypertensive disorders of pregnancy, offering valuable insights that are both population-specific and methodologically innovative.

Our findings support the notion that EOP and LOP are pathophysiologically distinct entities, and suggest that early identification of women at risk may be enhanced through personalized predictive models. Moreover, we identified key non-invasive predictors—including educational level, family history of perinatal death, number of sexual partners, primipaternity, and family history of hypertension—that could be used in low-resource settings to inform prenatal care decisions.

The analysis of the relationship between the factors under study and other adverse outcomes in pregnant women with preeclampsia yielded results that require further exploration. Future research should focus on expanding these models with larger multicenter cohorts and incorporating biomarker data (e.g., angiogenic factors, placental RNAs), genetic variants, and environmental exposures. In addition, the development of clinically applicable decision-support tools based on the best-performing algorithms may enhance risk stratification and timely interventions. Prospective validation and integration into electronic health records would be the next steps toward translational impact.

## Figures and Tables

**Figure 1 biomedicines-13-01958-f001:**
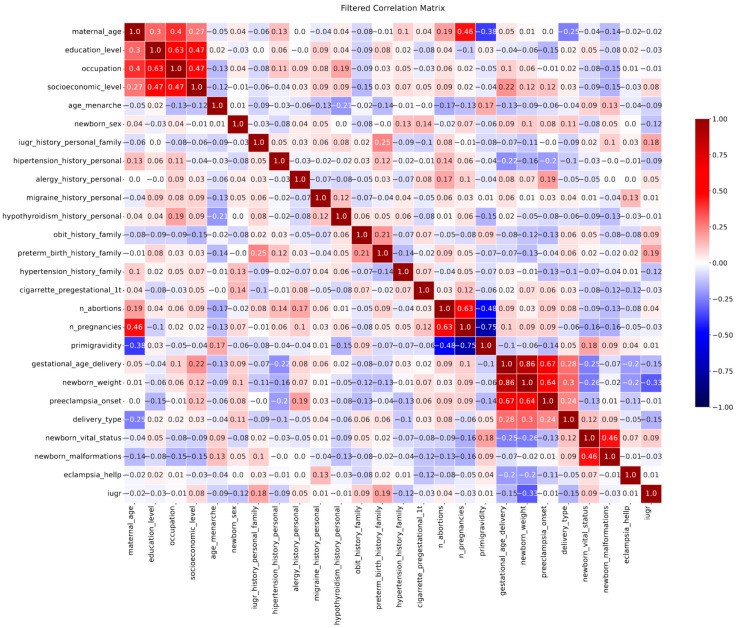
Filtered correlation matrix between particular risk factors and outcomes considered in this study. This matrix was obtained from the full correlation matrix (Appendix A) by applying a threshold of |ρ| ≥ 0.12 for the correlation coefficient.

**Figure 2 biomedicines-13-01958-f002:**
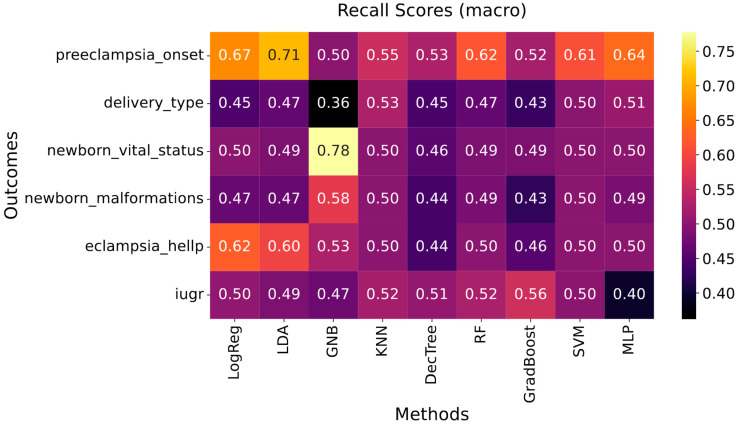
Recall of the machine learning models developed for the prediction of categorical outcomes, after variable selection. HELLP: hemolysis, elevated liver enzymes, low platelet count syndrome. IUGR: intrauterine growth restriction. LogReg: logistic regression; LDA: linear discriminant analysis. GNB: Gaussian naïve Bayes. KNN: k-nearest neighbors. DecTree: decision tree. RF: random forest. GradBoost: gradient boosting. SVM: support vector machine. MLP: simple multilayer perceptron.

**Figure 3 biomedicines-13-01958-f003:**
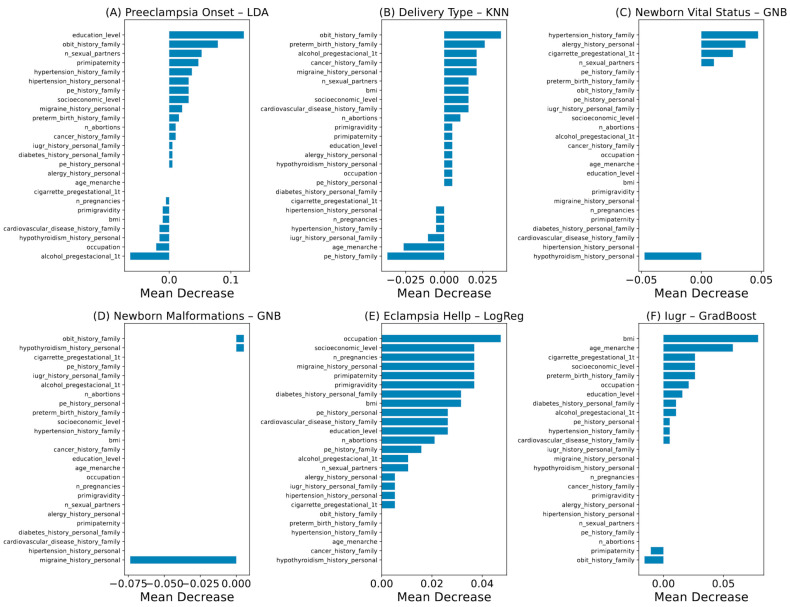
Contribution of variables for models that best predict the analyzed categorical outcomes. (**A**) PE onset. (**B**) Delivery type. (**C**) Newborn vital status. (**D**) Newborn malformations. (**E**) Eclampsia or HELLP syndrome. (**F**) IUGR. PE: preeclampsia. HELLP: hemolysis, elevated liver enzymes, low platelet count syndrome. IUGR: intrauterine growth restriction. LDA: linear discriminant analysis. KNN: k-nearest neighbors. GNB: Gaussian naïve Bayes. LogReg: logistic regression. RF: random forest.

**Table 1 biomedicines-13-01958-t001:** Population characteristics and distribution of risk factors and secondary outcomes.

Risk Factor or Secondary Outcome	Unit	EOP (*n* = 80)	LOP (*n* = 110)	*p* Value		All (*n* = 190)
Maternal age	years	27.50 (24–33)	28 (23–34.25)	0.8712	NS	28 (23–34)
Marital status	%			0.6876	NS	
Single, widowed, divorced, or separated		17.5 (14/80)	14.55 (16/110)			15.79 (30/190)
Married or cohabitating with the infant’s father		82.5 (66/80)	85.45 (94/110)			84.21 (160/190)
Education level	%			0.0420	*	
None		0.00 (0/80)	0.00 (0/110)			0 (0/190)
Incomplete primary education		0.00 (0/80)	2.73 (3/110)			1.579 (3/190)
Complete primary education		1.25 (1/80)	3.64 (4/110)			2.632 (5/190)
Incomplete secondary education		8.75 (7/80)	8.18 (9/110)			8.421 (16/190)
Complete secondary education		53.75 (43/80)	62.73 (69/110)			58.947 (112/190)
Incomplete university studies		8.75 (7/80)	3.64 (4/110)			5.789 (11/190)
Complete university studies		25.00 (20/80)	16.36 (18/110)			20 (38/190)
Postgraduate		2.50 (2/80)	2.72 (3/110)			2.632 (5/190)
Age at menarche	years	13 (12–14)	13 (12–14)	0.5277	NS	13 (12–14)
Sex of newborn	%			0.3032	NS	
Male		57.5 (46/80)	49.1 (54/110)			52.632 (100/190)
Female		42.5 (34/80)	50.9 (56/110)			47.368 (90/190)
Occupation	%			0.5194	NS	
Housewife		28.75 (23/80)	29.09 (32/110)			28.947 (55/190)
Student		7.5 (6/80)	4.54 (5/110)			5.789 (11/190)
Non-qualified technician		8.75 (7/80)	16.36 (18/110)			13.158 (25/190)
Qualified technician		32.5 (26/80)	26.36 (29/110)			28.947 (55/190)
Independent		2.5 (2/80)	5.45 (6/110)			4.211 (8/190)
Executive professional		20 (16/80)	19.18 (20/110)			18.947 (36/190)
BMI	Kg/m^2^	25.18 (22.50–27.89)	24.32 (22.63–27.52)	0.5375	NS	24.45 (22.63–27.66)
Personal PE background	%	10.0 (8/80)	10.0 (11/110)	>0.999	NS	10 (19/190)
Family history of PE	%	31.25 (25/80)	20.91 (23/110)	0.1284	NS	25.263 (48/190)
Personal or family history of IUGR	%	7.50 (6/80)	7.27 (8/110)	>0.999	NS	7.368 (14/190)
Personal history of chronic hypertension	%	23.75 (19/80)	9.09 (10/110)	0.0075	**	15.263 (29/190)
Personal history of allergy	%	0 (0/80)	8.18 (9/110)	0.0110	*	4.737 (9/190)
Personal history of migraine	%	7.5 (6/80)	9.09 (10/110)	0.7951	NS	8.421 (16/190)
Personal history of hypothyroidism	%	15 (12/80)	10 (11/110)	0.3688	NS	12.105 (23/190)
Family history of cardiovascular disease	%	17.5 (14/80)	19.09 (21/110)	0.8509	NS	18.421 (35/190)
Family history of spontaneous abortion	%	12.5 (10/80)	14.55 (16/110)	0.8313	NS	13.684 (26/190)
Family history of obit or perinatal death	%	8.75 (7/80)	2.73 (3/110)	0.0983	NS	5.263 (10/190)
Family history of preterm birth	%	18.75 (15/80)	15.45 (17/110)	0.5618	NS	16.842 (32/190)
Personal and family history of diabetes	%	36.25 (29/80)	32.73 (36/110)	0.6443	NS	34.211 (65/190)
Family history of cancer	%	12.5 (10/80)	20 (22/110)	0.2386	NS	16.842 (32/190)
Family history of hypertension	%	12.5 (10/80)	5.45 (6/110)	0.1124	NS	8.421 (16/190)
Pre-pregnancy and first trimester cigarette exposure	%	8.75 (7/80)	12.73 (14/110)	0.4849	NS	11.053 (21/190)
Pre-pregnancy and first trimester alcohol consumption	%	17.50 (14/80)	14.55 (16/110)	0.6876	NS	15.789 (30/190)
Primigravidity	%	50 (40/80)	36.36 (40/110)	0.0743	NS	42.105 (80/190)
Primipaternity	%	65 (52/80)	53.64 (59/110)	0.1368	NS	58.421 (11/190)
Number of abortions		0 (0–0)	0 (0–1)	0.1332	NS	0 (0–1)
Number of pregnancies		1.5 (1–2)	2 (1–2.25)	0.0904	NS	2 (1–2)
Number of sexual partners		1 (1–1)	1 (1–1)	0.1928	NS	1(1–1)
Socioeconomic status	%			0.0986	NS	
1 (lowest)		21.25 (17/80)	11.82 (13/110)			15.789 (30/190)
2		41.25 (33/80)	45.45 (50/110)			43.684 (83/190)
3		33.75 (27/80)	34.54 (38/110)			34.211 (65/190)
4		3.75 (3/80)	7.27 (8/110)			5.789 (11/190)
5 (highest)		0.00 (0/80)	0.91 (1/110)			0.526 (1/190)
Time in relationship with the infant’s father	Months	36 (18–93)	36 (14.5–108)	0.6951	NS	36 (18–96)
Gestational age at delivery	Weeks	31 (28.25–35.75)	37 (36–38.25)	<0.0001	****	36 (33–38)
Newborn weight	g	1418 (995–1974)	2668 (2259–3056)	<0.0001	****	2285 (1575–2880)
Vital status of the newborn	%			0.0983	NS	
Alive		91.25 (73/80)	97.27 (107/110)			94.736 (180/190)
Death		8.75 (7/80)	2.72 (3/110)			5.263 (10/190)
Type of delivery	%			0.0009	***	
C-section		91.25 (73/80)	71.81 (79/110)			80 (152/190)
Eutocic delivery		8.75 (7/80)	28.18 (31/110)			20 (38/190)
Malformations of the newborn	%	10 (8/80)	10.90 (12/110)	>0.9999	NS	10.526 (20/190)
IUGR	%	23.75 (19/80)	22.72 (25/110)	0.8638	NS	23.157 (44/190)
Eclampsia or HELLP	%	15 (12/80)	8.18 (9/110)	0.1631	NS	11.052 (21/190)

EOP: early-onset preeclampsia. LOP: late-onset preeclampsia. PE: preeclampsia. IUGR: intrauterine growth restriction. BMI: body mass index. HELLP: hemolysis, elevated liver enzymes, low platelet count syndrome. None of the quantitative variables analyzed had normal distribution. Quantitative variables with non-normal distribution are presented with their median (interquartile range). Qualitative variables are presented with their percentage (proportion). * *p* < 0.05; ** *p* < 0.01; *** *p* < 0.001; **** *p* < 0.0001.

**Table 2 biomedicines-13-01958-t002:** Statistically significant risk factors for pregnancy outcomes other than PE.

Risk Factor	Unit	Live Newborn (n = 180)	Stillbirth (n = 10)	*p*-Value	
Number of pregnancies		2 (1–2)	1 (1–1250)	0.0127	*
Primigravidity	%	40 (72/180)	80 (8/10)	0.0187	*
		**Without malformations (n = 170)**	**With malformations (n = 20)**		
Maternal age	Years	28 (24–34)	25.5 (22.25–28.75)	0.0491	*
Socioeconomic status	%			0.0359	*
1 (lowest)		13.529 (23/170)	35 (7/20)		
2		44.706 (76/170)	35 (7/20)		
3		34.706 (59/170)	30 (6/20)		
4		6.471 (11/170)	0 (0/20)		
5 (highest)		0.588 (1/170)	0 (0/20)		
Number of pregnancies		2 (1–2.25)	1 (1–2)	0.0461	*
		**Without IUGR (n = 146)**	**With IUGR (n = 44)**		
Personal or family history of IUGR	%	5 (7/146)	16 (7/44)	0.0211	*
Family history of preterm birth	%	13 (19/146)	30 (13/44)	0.0195	*
		**Cesarean (n = 152)**	**Vaginal delivery (n = 38)**		
Maternal age	Years	29 (24–35)	26 (21–30)	0.0010	***

PE: preeclampsia. IUGR: intrauterine growth restriction. None of the quantitative variables analyzed had normal distribution. Quantitative variables with non-normal distribution are presented with their median (interquartile range). Qualitative variables are presented with their percentage (proportion). * *p* < 0.05; *** *p* < 0.001.

## Data Availability

The original data presented in the study are openly available in GitHub at https://github.com/cfarkas/preeclampsia_ml (accessed on 30 July 2025).

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
