# Peer review of "Early-Onset Versus Late-Onset Preeclampsia in Bogotá, Colombia: Differential Risk Factor Identification and Evaluation Using Traditional Statistics and Machine Learning"

_biomedicines, 2025, doi:10.3390/biomedicines13081958_

Round 1

Reviewer 1 Report

Comments and Suggestions for Authors

The paper titled “Early-onset versus late-onset preeclampsia in Bogotá, Colombia” aims to identify and evaluate differential clinical risk factors associated with early-onset and late-onset preeclampsia in a Colombian population using both traditional statistical methods and machine learning techniques. 
This is a highly relevant and timely topic, especially considering the significant burden of hypertensive disorders in pregnancy across low- and middle-income countries. 
The study addresses a gap in the literature by focusing on a Latin American cohort—an underrepresented group in preeclampsia research—and aligns well with the scope of Biomedicines by integrating biomedical data with artificial intelligence approaches.

However, I have several concerns regarding the paper in its current form that warrant revision prior to acceptance:

MAJOR 1) 

The data were collected at a tertiary hospital in Bogotá between 2017 and 2018. 

Given this, what is the rationale for developing the system 8–9 years later?

MAJOR 2)
"The dataset was then splitted into training (80%) and external validation (20%)"

It contains 190 data in it.
20% means 38 test data
and recall means TP/(TP+FN)

80-110   38*(110/190)= about 22 positive data
So how recall of Decision Tree algorithms can be  0.53, 0.54, 0.55 for delivery type, iugr, preeclampsia_onset respectively?
I want to see the confusion matrix of the one of the algorithm.
It should be helpful for understanding the correctness of the used approach?

MAJOR 3)

There is a high correlation between delivery type and newborn weight (as 0.86)
So I have some doubts about the correctness of the Detection of the delivey type with 9 ML algorithms
The best one is 0.59
How this one can be possible. 
Please check these values in terms of recall formulation and number of data in your datset.
Or give your rationale for this?

MAJOR 4)
I want to learn whether the authors thinks about feature selection from this dataset.
Because depending on the definition of the features some of them directly seen as irrelevant with the target variable.
Therefore decreasing the number of features may result better values in the system.

MAJOR 5)
There are 38 features in the dataset
There are 190 data in the datset
110 of them are positive and 80 of them are negative
The authors used 6 of them as target variable.

I want ot learn how the authors seleted these 6 target features?
In the execution of ML system do they use these features as input variables or not?
Is it possible to measure these values with some devices? of by getting these information from the patient?
If so why do you use them as a target variable?

MAJOR 6)
were evaluated as predictors. Figure S1 and Table S1 present the predictive  -- ?? s1?
Why is it in the suplemenatry file?
Why is ti different from Figure2?

Some minor comments are as follows:

In "Table 1. Population characteristics and distribution of risk factors 242 and secondary outcomes ."
"15.79 (30/90)"  it should be 15.79 (30/190)
"NS 8.421 (16/190"-- it should be  NS 8.421 (16/190)

Author Response

We deeply thank the Reviewer for his/her careful evaluation of the submitted manuscript and useful comments. We have considered all of them in the preparation of this revised version. A detailed response to your comments is provided in the attached PDF.

Reviewer 2 Report

Comments and Suggestions for Authors

This paper gives a study on the Early-onset versus late-onset preeclampsia in Bogotá, Colombia. It studies the Differential risk factor identification and evaluation using traditional statistics and machine learning. After reviewing the paper, I have the following comments.
[1] The Population characteristics and distribution of risk factors and secondary outcomes are well described in Table 1.
[2] For the Correlation matrix between all the risk factors and outcomes considered in this study in Figure 1.  The fonts are quite small and difficult to read. The format and presentation of the matrix should be revised.
[3] How the machine learning model in 3.2 are selected and what are parameters for the training process? More details or flow chart for the process could be included in section 3.2 for better illustration.
[4] The fonts for the figure 2 should be enlarged for better presentation.
[5] The fonts for the figure 3 should also be enlarged for better presentation.
[6] The conclusion section 5 should be extended to describe more about the contributions of the paper and the implied future research direction.

Author Response

(The authors gave the same response as above.)

Reviewer 3 Report

Comments and Suggestions for Authors

Preeclampsia is a multisystem disorder. It is one main causes maternal and fetal morbidity and mortality in the world, especially when the condition is of early onset. Early diagnosis and prompt management are essential to preventing both maternal and neonatal complications through symptomatic management and delivery planning. Identification of differential risk factors between early- and late-onset preeclampsia, and using them for prediction is thought as important and actual direction in medicine researches. Authors try to assess the probability of address this issue using both traditional statistics and machine learning methods. Machine learning in medical statistics encompasses a broad spectrum of computational techniques that learn patterns from healthcare data to support clinical decision-making, risk prediction, and treatment optimization.

The study was conducted in accordance with the Declaration of Helsinki, and approved by the Ethics Committee. Informed consent for participation was obtained from all patients involved in the study.

There are some questions for authors.
1.    In the section 2.3. Traditional statistical analysis, the authors list the using methods. However, in the Results and the Tables there isn’t anything about which method is used for each parameter. What indicators had normality distribution? 
2.    The authors should be careful with description under Tables. So Table 1 has the note “Quantitative variables with normal distribution are presented with their mean ± standard deviation” but it doesn’t have such variables…
3.    The authors say “The group of women with EOP is characterized by having a higher education level” (line 236). Indeed, Table 1 has p = 0.042 for Education level and this value could be obtained if the authors apply only Chi-square test for trend. The Chi-squared test for trend is a useful tool for assessing relationships between two categorical variables and determining whether one variable increases or decreases as the other changes. The Chi-square test may not be reliable for small sample sizes, especially if expected frequencies are less than 5.
4.    The choice of validation methodology significantly affects the reliability of results. For small data sets, cross validation is preferred, and for large groups - separation into training and test sample. What is the reasoning behind the authors' choice of machine learning methods presented in the article?

Author Response

(The authors gave the same response as above.)

Round 2

Reviewer 1 Report

Comments and Suggestions for Authors

The authors made related corrections according to previous round of reviews.

It can be accepted as is.

Author Response

Thank you for your valuable comments and feedback.

Reviewer 3 Report

Comments and Suggestions for Authors

The authors have made extensive edits in accordance with the comments, rectifying some inaccuracies. However, in my opinion, the authors use an incorrect method to compare samples by level of education again. The Cochran-Armitage test for trend is a statistical method used to detect a linear trend in proportions across ordered groups. The objective is to determine if the proportion of binary outcomes in the ordered groups is increasing or decreasing in a monotonic way. The Cochran-Armitage test is not appropriate for situations where there are two samples (e.g., two groups). It's not a good idea to use the Cochran-Armitage exact test for trends if you're comparing two samples.
When there are 8 categories and two samples, the Fisher-Freeman-Halton test is the appropriate statistical tool to examine differences between the samples in terms of education level distribution. The exact Fisher-Freeman-Halton test for your groups, which excludes the row with only zeros, shows that p = 0.26.
From my perspective, the level of a mother's education is not linked to the outcomes.

In the Materials and Methods section (2.3 Traditional statistical analysis), the authors stated that the unpaired Student's t test was used to compare normally distributed numerical variables and summarised them as mean standard deviation. But I have not found variables that were summarised as mean ± standard deviation in the new version of the paper.
